# RADR: A Robust Domain-Adversarial-based Framework for Automated Diabetic Retinopathy Severity Classification

Sara Mínguez Monedero[*1]                     SARA.MINGUEZ.MONEDERO@STUDIUM.UNI-HAMBURG.DE
Fabian Westhaeusser[*2]                        FABIAN.WESTHAEUSSER@ZMNH.UNI-HAMBURG.DE
Ehsan Yaghoubi[1]                              EHSAN.YAGHOUBI@UNI-HAMBURG.DE
Simone Frintrop[1]                             SIMONE.FRINTROP@UNI-HAMBURG.DE
Marina Zimmermann[2]                           MARINA.ZIMMERMANN@ZMNH.UNI-HAMBURG.DE

[1] *Department of Informatics, University of Hamburg, Hamburg, Germany*

[2] *Institute of Medical Systems Biology, Center for Biomedical AI, Center for Molecular Neurobiology, University Medical Center Hamburg-Eppendorf, Hamburg, Germany*

**Editors:** Accepted for publication at MIDL 2024

## Abstract

Diabetic retinopathy (DR), a potentially vision-threatening condition, necessitates accurate diagnosis and staging, which deep-learning models can facilitate. However, in clinical practice these models often struggle with robustness due to distribution shifts caused by variations in data acquisition protocols and hardware. We propose **RADR**, a novel **R**obust domain-**A**dversarial-based deep-learning framework for **DR** severity classification, aimed at generalization across diverse datasets and fundus cameras. Our work builds upon existing research: we combine several ideas to perform extensive dataset curation, preprocessing, and enrichment with camera information. We then use a domain adversarial training regime, which encourages our model to extract features that are both task-relevant and invariant to domain shifts. We explore our framework in its various levels of complexity in combination with multiple data augmentation policies in an ablative fashion. Experimental results demonstrate the effectiveness of our proposed method, achieving competitive performance to multiple state-of-the-art models on three unseen external datasets.

**Keywords:** Robustness, domain generalization, adversarial training, diabetic retinopathy.

## 1. Introduction

Diabetic retinopathy (DR) is a medical condition that occurs due to microvascular retinal complications that are caused by diabetes mellitus. If the disease progresses, the result is irreversible vision loss, which is why early diagnosis of the disease is of utmost importance. DR is characterized by the presence of lesions in the eye: microaneurysms, hemorrhages, and soft and hard exudates. These are made visible on color fundus eye images and form the basis for evaluation of the severity of the disease by ophthalmologists (Wang and Lo, 2018; Lechner et al., 2017; Sun et al., 2022). According to the International Clinical Diabetic Retinopathy (ICDR) scale, five levels of severity can be defined: no DR, mild, moderate, severe, and proliferative (Wilkinson et al., 2003). Since the lesions are very small, manual diagnosis of DR is resource-intensive and time-consuming. This makes the development of algorithms to support the medical experts indispensable (Chen and Chang, 2022; Chetoui

---

[*] Contributed equally

and Akhloufi, 2020). Deep-learning (DL) models have achieved great success in various tasks in the field of medical image analysis, including DR grading (Li et al., 2021; Ragab et al., 2023; Litjens et al., 2017). However, real-world clinical data includes many sources for variations, such as imaging standards, camera brands, and patient demographics, which lead to different distributions or covariate shifts, like e.g. in the color space (Guan and Liu, 2021). As a consequence, DL models trained on one (or even several) specific domain(s) usually do not generalize well to other unseen domains, resulting in a lack of robustness for the application in real-world scenarios (Quinonero-Candela et al., 2008; Wang et al., 2022). Domain adaptation techniques, which aim at transferring knowledge from the source domain to the target domain or making models domain-agnostic, have emerged as a promising solution to this problem studied for various imaging modalities (Aubreville et al., 2023; Wang and Deng, 2018). Nonetheless, only limited research has been conducted in the field of domain adaptation for color fundus eye images (see Section 2). Therefore, the primary objective of this work is to develop a robust model for DR severity classification that generalizes to unseen domains. Our contributions include:

**DR severity classification framework:** We introduce **RADR**, a novel **R**obust domain-**A**dversarial-based DL framework for **DR** severity classification, aimed to generalize across diverse datasets and fundus cameras. This builds upon existing research in the field, by, to the best of our knowledge, combining for the first time extensive dataset curation based on quality control labels provided by Fu et al. (2019), camera domain information provided by Yang et al. (2020), data preprocessing as well as domain adversarial training and data augmentation.

**Ablation study:** We qualitatively and quantitatively assess the impact of various commonly used methods for domain generalization on our curated training dataset, namely domain adversarial learning, multi-camera (MC) training as well as AugMix and custom color augmentations, providing novel insights into their effectiveness in this field.

**Comparative analysis:** We finally conduct a comparative analysis with multiple state-of-the-art models, demonstrating the effectiveness of our proposed method in achieving performance comparable to or surpassing existing approaches on three unseen external datasets.

## 2. Related Work

To the best of our knowledge, four distinct approaches for domain adaption in the field of DR grading have been published to date. First, Yang et al. (2020) introduced the Residual-CycleGAN model for image-to-image translation. They defined camera labels for the images in the EyePACS dataset and used those cameras as domains. Adapted test images improved the performance of a classifier trained on a single camera, without retraining it on the target camera. However, this method requires retraining the CycleGAN for each additional domain. Additionally, they compared against a domain adversarial training strategy on camera labels, similar to what is used in our work, though without evaluating on external data or comparing to the SOTA. The main idea presented by Atwany and Yaqub (2022) is to use domain gradient variance information as a regularization technique. In addition, during training of their proposed DRGen model, they search for flat minima in the vali-

dation loss. For this purpose, their approach combines the Fishr method and stochastic weight averaging densely. Galappaththige et al. (2024) propose SPSD-ViT, which combines self-distillation with a prediction softening mechanism in vision transformers. Furthermore, their study involved retraining existing methodologies on a range of datasets, thereby facilitating a comprehensive comparative analysis of various SOTA model results. Finally, the multi-model learning approach presented by Zhang et al. (2022) treats each sample as a composite derived from multiple source domains. They combine models trained on different source domains, determining the model weight based on the Euclidean distance between the source and target models' features. Pseudo-labels for target images are obtained through feature-level clustering. They achieve over 90% accuracy on unseen data, though they transformed the five stages of DR into a simpler binary classification problem. Our work aims to expand this limited field of research, and wherever possible, benchmark against the existing methods.

## 3. RADR

In this work, we present RADR, a domain adversarial-based framework for automated diabetic retinopathy severity classification aiming for robust performance on unseen datasets. RADR is derived from the publicly available EyePACS dataset that we preprocess, curate for image quality and enrich with camera information. Using the camera labels as domains, we train RADR in an adversarial fashion to extract domain-agnostic features for its severity prediction. We finally compare our framework on three unseen commonly used DR datasets to the SOTA and evaluate our training regime in an ablative fashion. Figure 1 depicts the full pipeline. In the following, our approach is described in more detail.

### 3.1. Data

We train our model on the EyePACS dataset (Dugas et al., 2015), a publicly available collection of 88,702 color fundus eye images. These images are classified into five classes, corresponding to the level of DR severity. We apply multiple steps of preprocessing and data curation to this dataset. Firstly, roughly 25% of the images in the EyePACS collection are considered to be ungradable due to the poor quality, artifacts, excessively bright or dark images, or out-of-focus images (Chetoui and Akhloufi, 2020). Therefore, these poor quality images were eliminated according to the three-level quality labels provided by Fu et al. (2019), removing the 'reject' category. Additionally, overly dark images were removed by thresholding on the average pixel value of images converted to grayscale. All images are cropped and resized into squares of size $512 \times 512$ pixels, centered around the retina, to remove noise and redundant image areas. Furthermore, the images of the EyePACS dataset were acquired using different camera brands. To utilize this inherent known heterogeneity in our model, we use the camera labels provided by Yang et al. (2020) to create a separate subdataset for each camera A, B, C, D and E. For aggregated RGB histograms per subdataset, refer to the Appendix. It should be noted that they stated differences between the provided labels and those employed in their original approach. After applying curation and preprocessing, in total 62,467 images remain of the EyePACS dataset. We split every camera subdataset by 70/15/15% into train, validation and test sets, stratified by the severity label to assure equal distribution. In addition to the five camera datasets (A-E), three of

the most popular publicly available DR grading datasets are used as external datasets to evaluate the performance on unseen data and distributions. Those datasets are Messidor, Messidor 2 and APTOS (Abràmoff et al., 2013; Karthik, 2019). This will also allow us to compare the results obtained in this paper with those obtained by the SOTA models presented by Atwany and Yaqub (2022) and Galappaththige et al. (2024).

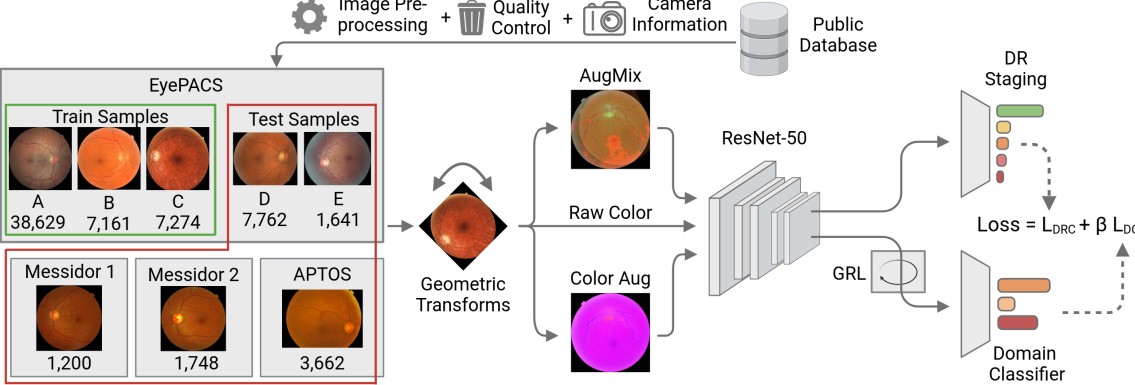

Figure 1: Methodology of the proposed RADR framework for DR severity classification. The EyePACS dataset is enriched by camera labels, curated for image quality and preprocessed. Subdatasets A, B and C are used for end-to-end model training. Images are augmented by geometric transformations, as well as optional ColorJitter or AugMix transformations, before being fed through a ResNet50-based encoder. Resulting latent features are forwarded to both the DR staging head and the domain classifier. The GRL inverts the sign of the gradient of the domain classification loss $L_{DC}$ in the encoder, promoting domain-invariant feature representations.

## 3.2. Model Architecture

In our pursuit to build a robust DR severity classification model, we first derive a single-camera (**SC**) baseline model comprised of a ResNet-50-based feature extractor (He et al., 2016) and a fully connected DR classifier with 5 output nodes. The SC model is trained only on camera A (source) and tested on each of the remaining subdatasets individually. In addition, geometric transformations are applied when training the SC model to increase data variability. These correspond to rotation, vertical and horizontal flipping and cropping. In the next step, we perform multi-camera (**MC**) training on the subdatasets from camera A, B and C. This is expected to increase the data variability that the model sees during training and therefore to increase performance on the remaining unseen datasets from camera D and E. In the last step, inspired by Ganin et al. (2016), we add a domain adversarial (**DA**) head to our network, composed of a Gradient Reversal Layer (GRL) and the domain classifier. This represents our final, most advanced model RADR. The domain classifier comprises two identical blocks, each with a linear layer, ReLu activation and dropout with probability of 0.5, followed by a final fully connected layer with three output nodes for domain classification. During backpropagation, the GRL inverts the sign of the gradient flowing from the domain head into the feature encoder (FE). Through parallel training

of both classification heads, weights of the FE are adapted to produce domain-agnostic features that are still predictive for the main task of severity classification. Both the DR classifier and the domain classifier are optimized using the AdamW optimizer and Cross-Entropy loss. By utilizing both the severity labels and the domain information of all three training subdatasets in parallel during training, our framework diverges slightly from the original DANN implementation, which only used the labels of a single domain for training the main task. Additionally, a hyperparameter $\beta$ was introduced to modulate the domain classification loss's impact. Therefore, the total loss $L$ of the model can be calculated as $L = L_{DRC} + \beta \cdot L_{DC}$ where $L_{DRC}$ corresponds to the loss computed by the DR classifier and $L_{DC}$ to the loss computed by the domain classifier. $\beta$ was empirically set to 0.3. Higher values tended towards degrading main task performance, while at lower values features could still be differentiated by the domain discriminator. We further wanted to analyze the influence of random data augmentations on the robustness of our models. For this, in addition to the default geometric transforms, we test the application of two different augmentation policies to the three presented levels of complexities of our pipeline, SC, MC, and DA training. The first policy employs color transformations, which randomly adjust the brightness, contrast, and saturation parameters of the images using ColorJitter of PyTorch's torchvision transforms (Paszke et al., 2019). As our second policy, we evaluate AugMix from Hendrycks et al. (2019), which is specifically designed to increase model robustness. In detail, AugMix creates multiple copies of an image, applies a unique data augmentation chain to each and then linearly combines them using random weights. For exact settings used in this work, refer to the Appendix. Finally, to evaluate the results obtained in classifying DR images, both quantitative and qualitative metrics are used. In this work, accuracy (ACC) and quadratic weighted kappa (QWK) are used as quantitative measurements, aligning with standard metrics in the literature to facilitate comparative analysis of the outcomes. We further utilize uniform manifold approximation and projection (UMAP) (McInnes et al., 2018) plots to visualize the latent representations before the final classification layer of all internal and external test data for qualitative analysis of the model.

## 4. Experiments

### 4.1. Quantitative Results: Internal Datasets

Table 1 depicts the performance in terms of QWK of all training regimes and data augmentation policies on the EyePACS camera subdatasets. For the corresponding ACC results, please refer to the Appendix. All variations of the model were trained end-to-end using AdamW optimizer with hyperparameters tuned individually for best QWK on the validation splits of the EyePACS camera subdatasets. When comparing the three training regimes with only their default geometric augmentation and no added color augmentation, a consistent increase in performance on both seen and unseen datasets can be observed when using the MC approach over SC approach. Average performance is further enhanced from 74.2% to 76.2% by applying domain adversarial (DA) training. On camera E, a slight drop is visible, though on camera D a major improvement of 8.1 percentage points could be achieved, emphasizing the robustness-conferring influence of the adversarial training on unseen domains. Remarkably, DA training also increased QWK on the cameras A, B & C seen during training, hinting at the possibility that the multi-task approach promoted

Table 1: Performance in terms of quadratic weighted kappa (QWK) (mean ± standard deviation) of our models on the test sets of the camera domains in the EyePACS dataset, trained with five different random seeds. SC: Single-camera training on camera A, MC: Multi-camera training on cameras A, B and C, DA: Domain adversarial training on cameras A, B and C. Best performing model in bold, second best underlined.

| QWK [%] | Camera A | Camera B | Camera C | Camera D | Camera E | Avg |
|---------|----------|----------|----------|----------|----------|-----|
| SC | 76.3±0.9 | 80.4±0.1 | 64.6±2.3 | 49.2±4.3 | 72.4±2.1 | 68.6 |
| SC ColorAug | 73.0±1.3 | 78.0±1.9 | 64.4±3.2 | 40.3±7.7 | 64.3±5.6 | 64.0 |
| SC AugMix | 74.1±0.8 | 75.8±1.4 | 57.0±4.4 | 34.6±2.4 | 58.4±2.7 | 60.0 |
| MC | 77.4±1.8 | 82.6±1.1 | 77.8±2.6 | 58.9±6.2 | 74.1±4.2 | 74.2 |
| MC ColorAug | 71.6±2.5 | 79.2±2.6 | 71.0±6.3 | 52.2±9.9 | 67.8±1.0 | 68.4 |
| MC AugMix | 76.3±0.9 | **85.3±1.6** | 77.6±1.1 | **68.6±6.0** | 72.2±1.5 | 76.0 |
| DA (RADR) | **78.1±1.8** | 84.4±0.6 | **78.7±1.0** | 67.0±4.2 | 72.6±1.6 | **76.2** |
| DA ColorAug | 73.1±3.3 | 80.9±2.3 | 74.7±3.9 | 48.8±1.6 | **74.5±6.1** | 70.4 |
| DA AugMix | 75.5±0.5 | 84.1±0.5 | 76.1±1.5 | 61.9±1.7 | 71.4±3.1 | 73.8 |

extraction of more predictive features in general. When applying color augmentations based on ColorJitter, results deteriorate for all training regimes and cameras, except for camera E under the DA regime. Here, in contrast, camera D performance dropped by 18.2 percentage points. Overall, this indicates that the ColoJitter augmentations evaluated by us contribute negatively to overall performance and robustness. Finally, the application of the AugMix policy decreased average performance under the SC and DA training regime, though increasing average performance under the MC training regime by 1.8 percentage points, achieving comparable performance to our proposed main model RADR. A potential explanation for the negative influence of the augmentations under the DA training regime is that it opposes the domain discrimination task by blurring the differences between camera domains, limiting the potential benefit of the domain adversarial training. However, further research is required to verify this.

### 4.2. Quantitative Results: External Datasets and SOTA

We evaluate our top performing model RADR, which employs domain adversarial training without added color transformations, as well as our second best performing approach using multi-camera (MC) training with AugMix augmentations on three unseen public DR datasets, Messidor 1, Messidor 2 and APTOS (Table 2). Here, we also analyze classification accuracy besides QWK to enable a comparison to existing SOTA methods. Of our models, RADR achieves the highest average QWK and accuracy with 76.7% and 65.7%, respectively, though MC training with AugMix performs slightly better on the APTOS dataset. Notably, the highly similar scores of RADR on the QWK metric indicate consistent and robust performance across all unseen datasets. For benchmarking our method against SOTA, we compare against models from Atwany and Yaqub (2022) and Galappaththige et al. (2024). We differentiate between approaches only using EyePACS as their

single source (SS) training dataset and those utilizing a multi-source (MS) leave-one-out training regime by training on the remaining three datasets when predicting on one out of EyePACS, Messidor 1, Messidor 2 and APTOS.

Table 2: Performance (mean ± standard devation) of our top-performing models, MC Aug-Mix and RADR, on the external datasets, trained with five different random seeds. SS: Single-Source training on EyePACS. MS: Multi-Source training in leave-one-out fashion on EyePACS, Messidor 1 & 2, as well as APTOS, with prediction on the remaining dataset. Best performing model in bold, second best underlined.

| **QWK** [%] | Messidor 1 | Messidor 2 | APTOS | Avg |
|---|---|---|---|---|
| **RADR** (Ours) | **77.7±1.9** | **75.2±2.7** | 77.2±2.1 | 76.7 |
| MC AugMix (Ours) | 76.1±1.3 | 71.4±7.0 | **81.9±1.8** | 76.5 |
| **ACC** [%] | Messidor 1 | Messidor 2 | APTOS | Avg |
| SS: **RADR** (Ours) | 65.3±1.3 | 71.6±2.2 | 60.2±2.9 | 65.7 |
| SS: MC AugMix (Ours) | 62.8±2.0 | 69.8±4.4 | 62.6±1.4 | 65.1 |
| SS: SPSD-ViT (Galappaththige et al., 2024) | 50.5±0.8 | 62.2±0.4 | **75.1±0.5** | 62.5 |
| SS: DRGen (trained by Galappaththige et al. (2024)) | 54.6±1.5 | 65.4± 0.1 | 61.3±1.9 | 60.4 |
| MS: SPSD-ViT (Galappaththige et al., 2024) | 64.8±0.5 | **72.4±0.6** | 51.7±1.2 | 62.9 |
| MS: DANN (trained by Galappaththige et al. (2024)) | 57.0±1.1 | 58.6±1.7 | 54.4±0.8 | 56.7 |
| MS: DRGen (trained by Galappaththige et al. (2024)) | 59.1±1.8 | 65.2±0.6 | 51.2±2.1 | 58.5 |
| MS: DRGen (Atwany and Yaqub, 2022) | **66.7** | 70.5 | 70.3 | **69.1** |

Here, the original DRGen method from Atwany and Yaqub (2022) achieved the best average performance with 69.1%. Our proposed RADR model scored second best with 65.7%. It is to note though that this comparison favors the DRGen model. By employing a leave-one-out training and evaluation regime on the four datasets EyePACS, Messidor 1&2 and APTOS, they not only utilized significantly more training data than us, the reported accuracies per unseen dataset also stem from different versions of their model, while our results are all from the same version. When aiming for generalization and robustness, reporting results from the same model across all unseen datasets should be preferred. Galappaththige et al. (2024) reproduced the DRGen method under the MS training regime, however, only achieved an average accuracy of 58.5%. Interestingly, they also evaluated a domain-adversarial network (DANN), similar to our approach, under the MS training regime. This achieved an average accuracy of 56.7%, lacking behind our method by 9 percentage points, even though they utilized more training data and multiple instances of their model. This hints at the superiority of utilizing the camera labels from Yang et al. (2020) as domain indicator for adversarial training, as in our proposed method, over defining every dataset as an individual domain. Finally, fair comparisons can only be drawn when comparing methods under the same SS training regime, only training on the EyePACS dataset and predicting on all others. Here, RADR outperformed the SPSD-ViT from Galappaththige et al. (2024) by 3.2 percentage points, as well as an SS re-implementation of the DRGen model by 5.3 percentage points. This concludes that our proposed framework is able to strongly compete with SOTA models, even by using less training data, or even surpass them under equal conditions.

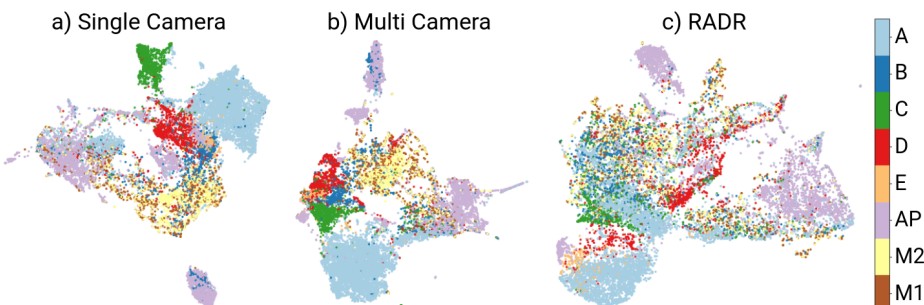

Figure 2: UMAPs for the camera subdatasets of EyePACS and external datasets for our three training regimes: Single Camera (a), Multi Camera (b) and RADR (c).

### 4.3. Qualitative Results: UMAP Representations

Figure 2 depicts UMAP visualizations of the latent representations in the output of the feature extractor of all samples across all datasets used in this work. For the SC (a) and MC (b) training regime, a high separability of the domain clusters of the camera subdatasets can be observed, especially for cameras A, C & D. This shows that for these approaches a lot of domain specific information is still contained after the FE, which isundesirable when aiming for robustness. When analyzing the external datasets, we observe that data from the same origin, Messidor 1 & 2, forms a mixed cluster, while APTOS is separate. The UMAP of the DA training regime of RADR (c) reveals a more entangled latent space, with the different domains blending into each other. This is especially the case for the EyePACS camera subdomains and the Messidor data, though APTOS still expresses a high separability from the remaining data. Overall, the visualization of the latent spaces emphasizes the successful push towards domain-invariant feature representations and robustness of our method.

### 5. Conclusion

This paper presented RADR, a deep-learning framework for DR severity classification, which combines several ideas to perform extensive dataset curation, preprocessing, and enrichment with fundus camera information with a domain adversarial training regime. We explored our framework in its various levels of complexity in combination with multiple data augmentation policies in an ablative fashion, showing best performance when only using geometric transforms during training. Our model achieved competitive or higher performance to multiple SOTA models on three unseen external datasets, even when using less training data. We link this mostly to the reduction of noise in the dataset by the extensive preprocessing and filtering we conduct, which has, to the best of our knowledge, never been done before by similar approaches aiming for robustness in DR classification. We hypothesize that further improvements could be achieved by retraining our model on all five camera domains instead of only three, as well as by employing more advanced data augmentations, specifically aimed to function in unison with a domain adversarial training regime.

## Acknowledgments

We thank Patrick Fuhlert, MSc, and Nico Kaiser, MSc, for their helpful comments.

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

## Appendix A. Histograms of EyePACS Camera Datasets

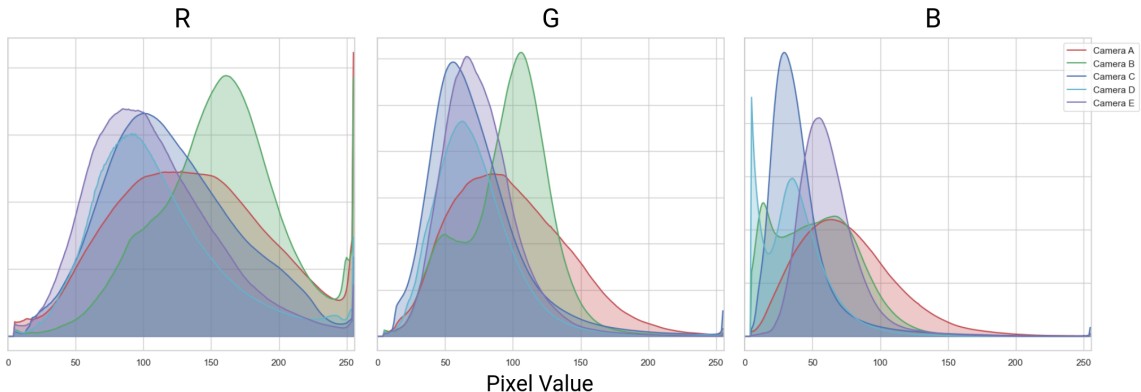

Figure A: RGB histograms for aggregated pixels of all images of the EyePACS dataset, separated by camera label. To filter out the spikes of the black pixels at the edges, all pixel values $< 5$ where removed.

## Appendix B. Model Parameters

Training of the SD, MD and DA model was performed for a maximum of 100 epochs with a batch size of 16. We used AdamW optimizer with a weight decay of 0.0005 and an initial learning rate of 1e-4. ReduceLROnPlateau scheduler was used with a reduction factor 0.2 and patience of 5 epochs. Early stopping is applied with a patience of 10 epochs. Dropout probability was set to 0.5. When adding the domain classifier, different learning rates were explored, however, it was found that using the same starting learning rate of 1e-4 lead to the best results. Furthermore, we also explored different values for $\beta$ between 0.1 and 1, with a final value of 0.3. Figure B depicts the architecture of the fully connected domain classifier. For the AugMix augmentation policy, both the severity of base augmentation operators and the number of augmentation chains was set to 3. The stochastic depth of augmentation chains was set to -1 and alpha to 0.1. The ColorAug policy was implemented using ColorJitter as provided by the Torchvision Python package. Adaptation ranges for brightness, contrast, and saturation were set to a range of [-0.3, 0.3]. In total, the ResNet50-based feature extractor contains 23.5M trainable parameters, the DR severity classifier 10.2k parameters and the domain classifier 3.1M parameters.

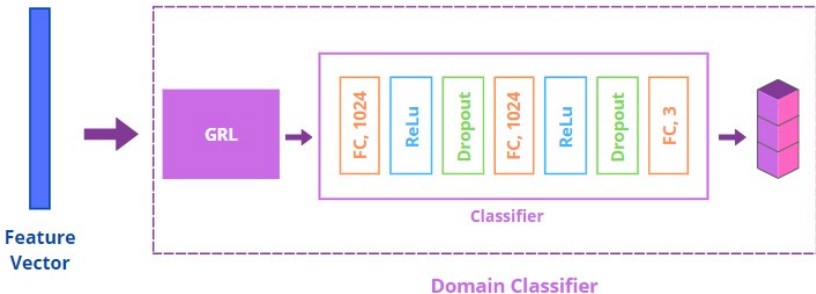

Figure B: Domain Classifier architecture.

## Appendix C. Extension Quantitative Results: Internal datasets

Table A: Performance in terms of accuracy (ACC) (mean ± standard deviation) of our models on the test sets of the camera domains in the EyePACS dataset, trained with five different random seeds. SC: Single-camera training on camera A, MC: Multi-camera training on cameras A, B and C, DA: Domain adversarial training on cameras A, B and C. All values are percentages. Best performing model in bold, second best underlined.

| ACC [%] | Camera A | Camera B | Camera C | Camera D | Camera E | Avg |
|---|---|---|---|---|---|---|
| SC | 75.2±1.1 | 80.8±1.1 | 77.7±2.1 | 67.2±4.5 | 81.7±0.8 | 76.5 |
| SC ColorAug | 78.7±1.1 | 81.1±1.4 | 78.4±1.9 | 63.3±7.3 | 80.0±3.3 | 76.3 |
| SC AugMix | 78.9±0.6 | 78.3±1.2 | 72.3±2.0 | 53.9±3.4 | 74.9±1.7 | 71.7 |
| MC | 79.2±4.1 | 81.5±2.0 | 83.3±3.2 | 67.8±2.7 | 82.5±2.8 | 78.8 |
| MC ColorAug | 73.0±7.3 | 77.2±3.5 | 75.2±1.5 | 59.1±1.6 | 76.4±5.7 | 72.2 |
| MC AugMix | **82.1±1.7** | **85.3±1.1** | 85.1±1.1 | **79.7±4.6** | 83.1±1.1 | **83** |
| DA (RADR) | 81.5±2.1 | 84.1±1.7 | **85.4±0.8** | 79.1±5.0 | **83.6±1.4** | 82.7 |
| DA ColorAug | 74.5±8.0 | 80.2±4.4 | 80.7±5.5 | 60.5±1.3 | 78.1±6.1 | 74.8 |
| DA AugMix | 81.2±0.4 | 84.7±0.3 | 84.1±0.8 | 75.4±1.6 | 82.7±1.3 | 81.6 |

