# OpenReview forum: "RADR: A Robust Domain-Adversarial-based Framework for Automated Diabetic Retinopathy Severity Classification"
_MIDL.io/2024/Conference — MIDL 2024 Poster_

### Official Review · Reviewer_n9ci · 2024-02-27

**Confidence:** 4
**Preliminary Rating:** 3
**Final Rating:** 4

**Summary:**

A novel Robust domain-Adversarial-based deep-learning framework for DR severity classification was built upon existing research. Qualitatively and quantitatively assessments of the impact of various commonly used methods for domain generalization on curated training dataset.

**Strengths:**

1. Comprehensive and robust experiments in both qualitative and quantitative.
2. Writings are good and clear, and contributions are summarized clearly.
3. Figures are well illustrated.
4. Comprehensive discussion of the reduction of noise in the conclusion.

**Weaknesses:**

1. The hyperparameter $\beta$ hasn't been discussed in the experiment.
2. The authors claimed that the DRGen model utilized more training data and stem different versions of models, but failed to explain/prove this statement.
3. The variable $k$ hasn't been defined or cited in the paper, please explain and discuss.
4. Baselines were not introduced in either the introduction or related works.

**Detailed Comments:**

The introduction of adversarial could be explained much clearer.

**Justification Of Final Rating:**

The authors have appropriately addressed the feedback to the manuscript and made modifications where needed, I appreciate the effort which they put in during such a short rebuttal period of time. The reviewer would like to recommend an acceptance based on the current manuscript.

**Justification Of The Preliminary Rating:**

As a validation paper, the experiments are somewhat sufficient and need a little supplement. The paper has clear writing and good illustrations but lacks of detailed explanation of the method. Meanwhile, there're several statements/hyperparameters not discussed enough throughout the paper. The reviewer is very likely to raise the score if the authors have addressed the above issues.

**Questions To Address In The Rebuttal:**

Please see the weaknesses.

---

> ### Author Response · Authors · 2024-03-16
> **Rebuttal - Reviewer n9ci**
>
> Thank you for your valuable insight and feedback, we appreciate your remarks very much. We hope the following answers and changes can help to mitigate the highlighted weaknesses:
>
> About β - Thank you for pointing this out! We explored different values for β between 0.1 and 1. We empirically determined 0.3 as the best value for β in our use case. At higher values, the influence of the DA training became too strong, such that features also became uninformative for the main task. At lower values, the features still contained too much domain information, such that the domain discriminator could easily differentiate them. We added the following to the manuscript: “β was set empirically to 0.3. Higher values tended towards degrading main task performance, while at lower values features could still be differentiated by the domain discriminator.“ We additionally added a more detailed description of the used hyperparameters for augmentation and training to the Appendix.
>
> About the utilized training data - The authors of the DRGen model [Atwany et al. 2022 - “DRGen: Domain Generalization in Diabetic Retinopathy Classification”] evaluated their model’s performance under a leave-one-out regime on the four datasets Messidor, Messidor 2, APTOS and EyePACS. That means that their reported performance per test dataset always stems from training on the three remaining datasets (refer to Table 3 & 4 in their paper). Since the full EyePACs dataset is always in their training data for the reported external datasets Messidor, Messidor 2, APTOS in our paper, and our method only utilizes three out of five camera domains from EyePACS for training, the utilize significantly more training data than us. Moreover, since they retrain their model for every test dataset on the three other datasets, every reported accuracy stems from a different version of their model. However, our reported results on Messidor, Messidor 2, APTOS are all created by inference on the same RADR model. We changed the sentence in the manuscript to highlight this to: “By employing a leave-one-out training and evaluation regime on the four datasets EyePACS, Messidor 1&2 and APTOS, they not only utilized significantly more training data than us, the reported accuracies per unseen dataset also stem from different versions of their model, while our results are all from the same version.”
>
> About the variable 𝜅 - Thank you for mentioning that we should explain or introduce this variable. It was referring to the number of image copies that are created and augmented during the AugMix procedure, before being merged into a single augmented image. In our work, we used 3 copies, since this was the default setting of the official Pytorch implementation. We note that we did not tune this parameter due to time constraints and focus on other aspects of this work that we deemed more important. To avoid confusion, we removed the “k” from the main text. As mentioned, we now added all augmentation, training and model based hyperparameters to the Appendix.
>
> About the introduction of baselines - Thank you for this comment. Based on this, we updated the Related Work section to clearly introduce the baselines with their proposed model names (DRGen of Atwany and Yaqub (2022) and SPSD-ViT Galappaththige et al. (2024)) for easier referencing.
>
> About description of the adversarial training - Thank you for pointing this out. We modified Section 3.2 to describe the used adversarial approach clearer: “In the last step, inspired by Ganin et al. (2016), we add a domain adversarial (DA) head to our network, composed of a Gradient Reversal Layer (GRL) and the domain classifier. This represents our final, most advanced model RADR. The domain classifier comprises two identical blocks, each with a linear layer, ReLu activation, and dropout, followed by a final fully connected layer with three output nodes for domain classification. During backpropagation, the GRL inverts the sign of the gradient flowing from the domain head into the feature encoder (FE). Through parallel training of both classification heads, weights of the FE are adapted to produce domain-agnostic features that are still predictive for the main task of severity classification.”
>
> Changes in the manuscript:
>
> We added the exact setting for β of 0.3 to the Methods.
>
> We added a clearer description about the leave-one-out training/evaluation procedure of the DRGen model to the Experiments section.
>
> We added the names of the baselines architectures to the Related Work section.
>
> We added a more detailed description of the used hyperparameters for augmentation and training in the Appendix.
>
> We modified Section 3.2 to provide a clearer description of the adversarial approach.
>
> We added the sentence to encourage further research on the influence of data augmentation at the end of the Internal Results.
>
> We added aggregated RGB histograms for the camera datasets to the Appendix.

---

### Official Review · Reviewer_m9oV · 2024-02-28

**Confidence:** 4
**Preliminary Rating:** 4
**Recommendation:** Poster
**Final Rating:** 5

**Summary:**

This paper adopts adversarial training to improve the accuracy of diabetic retinopathy severity stage grading.
A main classification DL model processes fundus photographs and is coupled with a small classifier which is tasked to recognize the image domain (here camera labels). By using a reversal gradient layer the methods implements an adversarial form of training, which helps the main model in extracting features invariant to changes that may be specific to the camera. Combined with extensive data cleaning and augmentation, the approach results in enhanced model's robustness making it perform comparably to SOTA approaches on unseen publicly available datasets.

**Strengths:**

This is a very relevant paper, well written with rigorously conducted experiments.
It uses ideas from previous work such as adversarial learning of domain invariant features, but it introduces ideas not tested before such as the use of camera labels as the domain label for the adversarial task.
I appreciated also the ablation study and the performance comparison with other SOTA methods on publicly available datasets.

**Weaknesses:**

Data augmentation plays a key role in this paper and in some cases data augmentation appeared to have degraded performance.
I think the manuscript should provide more details about the data augmentation which could be a useful lesson on what worked and what did not work. Future researchers can start from there.

**Detailed Comments:**

- I am curious about the fact that data augmentation did not help boost performance, so would like to genuinely ask whether they quality checked the images post- augmentation and whether they have built an intuition on what might have affected performance when data augmentation is in place
- I would like to understand how correlated dataset A, B, C, D and E are to each other

**Justification Of Final Rating:**

I would like to thank the authors for their response, I appreciate the effort which they put during such short rebuttal period of time. I was already supportive with presenting this paper but would like to upgrade my rating to strong accept.

**Justification Of The Preliminary Rating:**

- The paper is relevant for this conference
- The topic is important
- The experimental study was well conducted
- The idea although is not original, contains aspects like the usage of camera labels which I believe researchers in the field will find interesting

**Questions To Address In The Rebuttal:**

I would like the authors to focus a moment on the limitation of the approach and how it could be improved.

**Special Issue:**

No

---

> ### Author Response · Authors · 2024-03-16
> **Rebuttal - Reviewer m9oV**
>
> Thank you for your valuable insight and feedback, we appreciate your remarks very much. We hope the following answers and changes can help to mitigate the highlighted weaknesses:
>
> About the data augmentation - Thank you for pointing this out, this is a very relevant question. We did quality check the images before to verify visually that the images did not deviate into “unrealistic” regimes. But given research on other datasets like ImageNet, where very strong color augmentations, even to the point of “unrealistic” looking images provided good generalization results [Cubuk et al. 2019 - “AutoAugment: Learning Augmentation Strategies from Data”], it is difficult to quantify visually beforehand whether an augmentation will have the desired effect. We added the exact hyperparameter settings for AugMix and ColorJitter, based on your remark, to the Appendix section and also added a remark pointing to the Appendix in the main text. To understand why augmentation did not provide the desired effects in some cases or even had detrimental influence is a very interesting topic that would, however, require further analysis. We added the sentence “However, further research is required to verify this.” at the end of the Internal Results section to encourage this. If we would need to formulate a suspicion for the detrimental effects, besides the one given in the manuscript (Augmentation opposes DA), it would be that the chosen augmentation ranges did not properly cover the domains of the unseen cameras and rather “confused” the network by introducing noise in the wrong “direction”.
>
> About the correlation of camera datasets - This is a very interesting question, thank you for that! Measuring “correlation” between datasets or domains is a key aspect in the field of domain generalization or robustness and not a straightforward question to answer in our opinion. We think that correlation could be measured on different factors, such as average color intensities per dataset, texture of the images, systematic geometric differences and so on. The most promising individual factor would probably be color distribution. For this, we calculated the aggregated RGB histograms for all camera datasets and attached the plot to the Appendix. Here, shifts in the red, blue and green channel between the camera datasets are visible.
>
> About the limitations of the approach and how it could be improved - Thank you for this question. We think that the most significant limitation of the proposed approach is that it requires data from multiple domains that express a sufficient domain shift which can be utilized by the adversarial training. Additionally, the domains need to be known and labeled in advance. If only a single domain is available for training, the method can not be applied. On the side of potential improvements it would be desirable to extract domains directly from the underlying data, similar to what the authors in (Marini et al. - “H&E-Adversarial Network: A Convolutional Neural Network To Learn Stain-Invariant Features Through Hematoxylin & Eosin Regression”) propose for histopathological images. Here, instead of defining the domains by acquisition protocol, they calculate the stain matrix for every image, which incorporates color information. They then perform adversarial regression on the stain matrix, such that basically every image provides its own “domain” that should be unlearned. A similar approach would be thinkable for fundus images, by performing adversarial regression on e.g. the hue channel of the images.
>
> Changes in the manuscript:
>
> We added the exact setting for β of 0.3 to the Methods.
>
> We added a clearer description about the leave-one-out training/evaluation procedure of the DRGen model to the Experiments section.
>
> We added the names of the baselines architectures to the Related Work section.
>
> We added a more detailed description of the used hyperparameters for augmentation and training in the Appendix.
>
> We modified Section 3.2 to provide a clearer description of the adversarial approach.
>
> We added the sentence to encourage further research on the influence of data augmentation at the end of the Internal Results.
>
> We added aggregated RGB histograms for the camera datasets to the Appendix.

---

### Official Review · Reviewer_uwCk · 2024-03-03

**Confidence:** 5
**Preliminary Rating:** 4
**Final Rating:** 4

**Summary:**

The authors propose a domain adversarial framework for developing a robust diabetic retinopathy classification that is invariant to data shifts due to scanners and institutional differences. The paper provides a strong comparison of their results with relevant SOTA methods, popular data augmentation methods, and ablation studies.

**Strengths:**

- The paper is nicely written and is easy to follow the approach and motivation for this research.
- Results of the method compared with commonly used data augmentations provide insights into how they could be negatively influencing model training while aiming for domain generalization. However, this needs further reproducibility for other tasks.
- Even though this approach of adding camera information is not entirely novel, its application on diabetic retinopathy classification might be useful to investigate scanner differences in datasets and room for harmonization of results.

**Weaknesses:**

- In Figure 2 which depicts UMAP visualizations of latent space; each dataset also contains different DR severity samples which seem to be clustered very closely. One would ideally expect the same severity samples to be clustered together invariant of domain/dataset. However, this is not visible from these plots.
- Authors should also mention model parameters of their model in comparison to included SOTA as it has been shown to matter when analyzing data augmentations and domain generalization.

**Detailed Comments:**

- Please also mention what percentage of dropout was used in the model architecture in training, as that is relevant to the application theme here.

**Justification Of Final Rating:**

The authors have appropriately addressed the feedback to the manuscript and made modifications where needed. Even though the ideal is not entirely novel, these results on data augmentations and discussions around it would be useful to other researchers for further analysis.

**Justification Of The Preliminary Rating:**

The idea of using camera/domain information for a robust classification model invariant to the domain has been investigated before. However, its application to  DR classification is useful for further research and downstream analysis.

**Questions To Address In The Rebuttal:**

It would be useful for the research community if authors could provide comments on above mentioned weaknesses.

---

> ### Author Response · Authors · 2024-03-16
> **Rebuttal - Reviewer uwCk**
>
> Thank you for your valuable insight and feedback, we appreciate your remarks very much. We hope the following answers and changes can help to mitigate the highlighted weaknesses:
>
> About the DR severity labels in the UMAPs - Thank you for this remark. UMAP is an unsupervised clustering approach that clusters data based on variance present in the data. You are absolutely correct that in an ideal setting, where the strongest separator in the latent data is the DR severity label, the UMAP would cluster based on this label. However, since there is other bias in data, like bright/dark images etc., this might still lead to a stronger unsupervised clustering than the actual severity labels (Though the subsequent classification layer is still perfectly able to separate the latent space based on the label). Therefore we wouldn’t expect that in our model the UMAP clusters are mainly based on the DR severity. However, we wanted to prove that the domain information as the main source of variance has been reduced between the SD and the DA model, such that the UMAP doesn’t cluster based on the domain as its “easiest solution”, which was achieved in our opinion. This insightful question got us thinking for a while and we are more than happy to dive deeper into this with the reviewer in the following discussion period.
>
> About the model parameters - Thank you for pointing out that we did not explain the hyperparameters and the number of trainable parameters in our model. We added a section to the Appendix covering all relevant hyperparameters for the models, the training and the augmentation policies. Also, the number of trainable parameters for the different parts of our model is included there. These refer to 23.5M for the ResNet50-based feature extractor, 10.2k for the DR severity classifier 3.1M for the domain classifier.
>
> About dropout - You are correct that this is a very relevant mention, especially when talking about generalizability. We used dropout with a probability of 0.5. We added this number to the relevant section in the main text, as well as to the newly introduced list of used hyperparameters in the Appendix.
>
> Changes in the manuscript:
>
> We added the exact setting for β of 0.3 to the Methods.
>
> We added a clearer description about the leave-one-out training/evaluation procedure of the DRGen model to the Experiments section.
>
> We added the names of the baselines architectures to the Related Work section.
>
> We added a more detailed description of the used hyperparameters for augmentation and training in the Appendix.
>
> We modified Section 3.2 to provide a clearer description of the adversarial approach.
>
> We added the sentence to encourage further research on the influence of data augmentation at the end of the Internal Results.
>
> We added aggregated RGB histograms for the camera datasets to the Appendix.

---

### Comment · Area_Chair_8Jde · 2024-03-15
**Gentle reminder to post a rebuttal**

Dear authors,

This is just a gentle reminder that the rebuttal deadline is coming up. I encourage you to take this opportunity so you can engage with the reviewers in the upcoming discussion phase.

Best wishes,
AC

---

### Meta-Review · Area_Chair_8Jde · 2024-04-02

**Recommendation:** Accept (Poster)
**Confidence:** 5

**Metareview:**

The reviewers agree this paper is relevant, well written, and comprehensively and rigorously evaluated. The reviewers appreciated the efforts the authors made during the rebuttal phase, which improved the paper and addressed the reviewers’ concerns.

I follow the consensus recommendation of the reviewers and recommend acceptance of the paper.

---

### Decision · Program_Chairs · 2024-04-06

Accept (Poster)